# CONTINUAL MEMORY NEURONS

## ABSTRACT

Learning with neural networks by continuously processing a stream of data is very related to the way humans learn from perceptual information. However, when data is not i.i.d., it is largely known that it is very hard to find a good trade-off between plasticity and stability, frequently resulting in catastrophic forgetting issues. In this paper we follow a novel route, tackling the problem at the lowest level of abstraction. We propose a neuron model, referred to as Continual Memory Neuron (CMN), which does not only compute a response to an input pattern, but also diversifies computations to preserve what was previously learned, while being plastic enough to adapt to new knowledge. The values attached to weights are computed as a function of the neuron input, which acts as a query in a key-value map, with the goal of selecting and blending a set of learnable memory units. We show that this computational scheme is motivated by and strongly related to the ones of popular models that perform computations relying on a set of samples stored in a memory buffer, including Kernel Machines and Transformers. Experiments on class-and-domain incremental streams processed in online and single-pass manner support CMNs' capability to mitigate forgetting, while keeping competitive or better performance with respect to continual learning methods that explicitly store and replay data over time.

## 1 INTRODUCTION AND RELATED WORK

Learning from a data stream which is continuously provided by a given source represents one of the most challenging learning settings for machines, despite it being very related to the way humans learn from perceptual information (Betti et al., 2022). Neural networks are known to suffer from catastrophic forgetting, being hard to find a good trade-off between plasticity and stability (Parisi et al., 2019). This problem is the subject of many scientific papers in the existing literature on continual/lifelong learning (Delange et al., 2021; Mai et al., 2022b). The variety of current existing approaches is significant (Delange et al., 2021; van de Ven et al., 2022), and many of them assume to work in very-specific experimental conditions, such as in problems in which data from different tasks is sequentially streamed in large batches (Li & Hoiem, 2017; Aljundi et al., 2018a), possibly having access to the task boundaries. When considering a general setting, it turns out that one of the most simple and effective strategies consists in buffering and appropriately re-using (during learning) a portion of the processed examples (Aljundi et al., 2019b; Zhang et al., 2022), either represented in the input space (Prabhu et al., 2020; Zhou et al., 2023) or in some latent spaces (Buzzega et al., 2020).

In this paper, we face continual learning in its most essential form: we simply consider a stream of data which, at each time instant $t$, yields a data sample $x^{(t)}$, possibly paired with some label information $y^{(t)}$. Samples are expected to be presented in a not i.i.d. manner, and no other information is available. Without any loss of generality, we focus on the most challenging setting, that is the one of continual single-pass online learning, where the model is updated after having processed each $(x^{(t)}, y^{(t)})$. Of course, we assume that storing data is not a viable option to solve the problem, since the data stream could be potentially infinite. Motivated by the aforementioned considerations on the performance of memory-buffer-based models (being them simple rehearsal strategies (Zhang et al., 2022) or structured architectures that also include memory buffers (Ermis et al., 2022; Wang et al., 2022)), we propose to follow a direction that is *orthogonal* to the existing literature, to the best of our knowledge. In particular, we rethink the way neural networks operate at a very low level, proposing a novel *neuron model*, referred to as Continual Memory Neuron (CMN) which includes a special type of learnable memory. A CMN is explicitly designed to learn from information coming from a

non-stationary distribution, allowing the network to perform almost isolated computations in different regions of the input space, thus mitigating catastrophic forgetting but still keeping the capability of being plastic (Fig. 1). CMNs are a generalization of classic neurons, where the values of the weights entering the neuron become function of its input, and are obtained by blending internal memory units. Hence, a CMN does not only compute a response, but it also internally selects the process that leads to the response, i.e., what is the appropriate set of weights to use in a certain region of the input space. The learnable memory units represent values in a key-value learnable map, where the neuron input acts as a query to the map. When restricted to a single memory unit, CMNs degenerate to classic neurons (Fig. 1). In principle, every existing neural network can gain continual learning skills by replacing the neuron model with the one of CMNs (even if such an analysis goes beyond the scope of this paper). Similarly, also existing continual learning approaches based on neural networks could be revisited by exploiting CMNs. As a result, the goal of this paper is not to propose a state-of-the-art continual learning strategy, but to validate a novel direction that might open important perspectives to the scientific community. We show how this computational scheme is inspired by models that are specifically designed to compute a response in function of data belonging to a memory buffer, such as Kernel Machines (Schölkopf et al., 2002; Gnecco et al., 2015) (training data), Transformers (Vaswani et al., 2017) (input tokens), and others. We evaluate CMNs in some carefully designed benchmarks and class-and-domain incremental problems, showing how CMNs can overcome common methods based on memory buffers and rehearsal, without storing or replaying past information.

In summary, our contributions: (1) we propose a novel neuron model with memory units, explicitly designed for continual learning purposes and more general than classic neurons; (2) we show the connections between the proposed scheme and well-established models in Machine Learning; (3) we experimentally validate the quality of CMN-based networks, without any attempts to beat the state-of-the-art, but with the aim of paving the way for future investigation in this very novel direction.

**Related Work.** In the context of continual learning, standard parameter isolation methods bypass task interference by allocating different parameters to each task (Aljundi et al., 2019c; Mai et al., 2022b), generally requiring task-related information to route the computations towards certain isolated model components, both in fixed and dynamic architectures (Delange et al., 2021). Some works leverage learned gating mechanisms implemented as autoencoders that route the computation toward different experts (Aljundi et al., 2017). Our work shares with *conditional computation* (Lin et al., 2019; Abati et al., 2020; Shazeer et al., 2017) the idea of making the parameters of the neural network a function of the input. Lin et al. (2019) proposed a clipped version of maxout nets with partial conditional computation. Parameters are partially shared among examples, with the purpose of locating a set of examples to be replayed that interfere with the currently processed sample. Differently, in our work we propose a general model of the neuron with a rich computational capability (keys and memory units) completely avoiding any kind of exemplar replay. Abati et al. (2020) deal with class-incremental continual learning by shifting the complexity from a class prediction to a task prediction level. The requirement of clear task distinction holds in order to gain any benefit from this architectural design. Conversely, our approach is general and completely agnostic of task boundaries. Out of the scope of continual learning, Shazeer et al. (2017) trained gating mechanisms by gradient descent, learning to route the computations towards different model components. Conversely, CMNs build a natural parameter isolation scheme by leveraging a query-key-value mechanism. Several works investigated the algorithmic side of continual learning keeping the architecture fixed (Mittal et al., 2021; Rolnick et al., 2019; Zhang et al., 2020; Zhou et al., 2022), or evaluating the role of standard architecture components (pooling, batch norm, etc.) and net structure (Mirzadeh et al., 2022a;b), for example highlighting the role of width and depth of the neural layers. CMNs follow a direction that is orthogonal to such literature, rethinking the way neural networks operate at a very low level, being them novel neuron models explicitly designed to learn from information coming from a non-stationary distribution. The key management of CMNs is also different from models that cluster the network-input data, strongly depending on class-label information to organize an internal storage of centroids (Ayub & Wagner, 2020). Ren et al. (2021) use a prototype-oriented memory module based on similarity to input and probabilistic clustering, with additional loss terms to favour cluster development. CMNs shares some intuitions with works based on winner-take-all strategies (or on top-$k$ sparsity (Bricken et al., 2023)) for continual learning (Aljundi et al., 2018b; Iyer et al., 2022). Srivastava et al. (2013) used a local version of top-$k$, defining disjoint subsets of neurons in each layer and applying top-$k$ locally to each. Recently (Bricken et al., 2023) introduced a number of modifications to MLPs resulting in a model capable of continual learning. While these method require

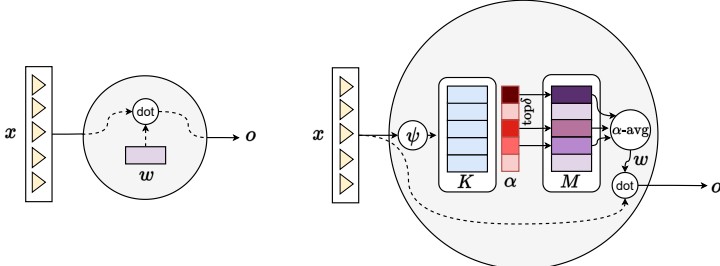

**Figure 1:** Classic neuron, computing the dot product between weights $w$ and input $x$. (a-right) Continual Memory Neuron, composed of learnable keys $K$ and memory units $M$, returning $w$ as the outcome of blending (weighted average–avg) multiple memories in function of the input $x$.

to condition the top-$k$ selection on an external signal (e.g., task-ID), our sparse routing mechanism is implicitly implemented by a query-key-value mechanism completely agnostic of task-related info.

## 2 NEURON MODEL

Each neuron in a neural network is an elementary unit computing a scalar value given an input $x \in \mathbb{R}^u$, $u > 0$. A neuron is characterized by $n$ learnable parameters $w \in \mathbb{R}^u$, a learnable scalar bias $b$, and a usually non-linear activation function $\sigma$. For the sake of simplicity, and without any loss of generality, we avoid explicitly indicating $\sigma$ and $b$, thus the output function of the neuron is

$$f(x, w) = w'x, \tag{1}$$

being $w'$ the transpose of the weight vector $w$. Such a vector defines the way the neuron responds to its input, and it is progressively developed during the learning process, i.e., it is the *memory unit* of the neuron (Fig. 1, left).

A Continual Memory Neuron (CMN) is a novel model of neuron, designed with the idea of augmenting the computational mechanism of Eq. 1 with increased memorization capabilities. It is motivated by the idea of making neurons better suited to deal with continual learning problems, where learning from the currently streamed data should not destroy what was learned in the past (i.e., avoid strong forgetting). In detail, a CMN generalizes the classic notion of neuron, introducing ($a$) a set of $m \geq 1$ learnable *memory units* that are appropriately blended to yield the weight vector $w$, and ($b$) an additional computational mechanism that determines what memories to blend (and how) in function of the neuron input $x$. In order to reach this goal, each memory unit is paired with a learnable $u$-sized key, and an attention mechanism compares $x$ with such keys, yielding stronger attention scores for keys whose memories will have a stronger contribution in defining the weight vector (Fig. 1, right). Formally, memory units are collected row-wise into the $m \times u$ matrix $M$, while the $m \times u$ matrix $K$ collects (row-wise) $m$ keys. Eq. 1 becomes

$$f(x, K, M) = \hat{w}(x, K, M)'x, \tag{2}$$

where function $\hat{w}$ returns a weight vector $\in \mathbb{R}^u$ depending on the current input $x$, keys and memories,

$$\hat{w}(x, K, M) = M'\alpha(x, K, \delta). \tag{3}$$

while $\alpha$ determines the relevance of each memory unit according to the similarity between $x$ and the keys in $K$ ($\delta$ will be described shortly). Notice the two-fold dependence on $x$ in Eq. 2. Function $\alpha$ returns a vector of $m$ positive (attention) scores that sum to one, and it is characterized by three main properties suitable for continual learning purposes. ($i.$) Attention scores are constrained to be sparse, with the parameter $\delta \in [1, m)$ defining the maximum number of non-zero entries. Sparsity is a crucial feature to implement a natural form of parameter isolation, that is typical of a large category of continual learning methods (Delange et al., 2021). In fact, it guarantees that the excluded $m - \delta$ memories will not be altered by the learning process, since the neuron output is not function of them, preserving the information they store and mitigating catastrophic forgetting. ($ii.$) Moreover, $\alpha$ is expected to return high values for keys "close" to $x$, in the sense inducted by the way such similarity function is implemented. This means that similar neuron inputs will trigger the same memory units and yield similar attention scores, while significantly different inputs will trigger different memories. Two inputs that are not close and not distant might trigger sets of memory units that intersect, giving the same or different emphasis to the shared units, thus favoring a transfer of information between nearby regions of the input space, which is another important property in continual learning. ($iii.$)

Ideally, $\alpha$ should be computationally simple in practical implementations. Additionally, it is easy to verify that CMNs formally extends vanilla neurons with augmented memory.

**Theorem 1.** *Neurons in artificial neural networks are CMNs with a single memory unit.*

*Proof.* When the number $m$ of memory units is 1, $\alpha$ necessarily returns 1 for all $x$ and $K$, due to the constraint of returning $m$ positive values that sum to 1, thus making $\hat{w}$ independent on $x$ and $K$. $M$ is a single-row matrix, thus $\hat{w}(x, K, M) = M'1$ is a column vector that corresponds to $w$ in Eq. 1, making Eq. 1 and Eq. 2 equivalent. $\qquad\square$

**Implementation.** Referring to the three aforementioned requirements $i.$, $ii.$, and $iii.$, we define $\alpha$ by means of $\text{softmax}^\delta$, which is the softmax function restricted to the top-$\delta$ logits, setting to 0 all the excluded components in the output of $\text{softmax}^\delta$, thus ensuring the $\delta$-sparsity requirement ($i.$),

$$\alpha(x, K, \delta) = \text{softmax}^\delta(\gamma \cdot \text{sim}(\psi(x), K)), \tag{4}$$

where $\gamma > 0$ is a temperature parameter that tunes its sharpness and $\text{sim}(\cdot)$ is a similarity function ($ii.$) comparing $\psi(x)$, which is a simplified representation of $x$, with the keys in $K$, and returning $m$ similarity scores. The function $\psi$ is a fixed transformation that maps $x$ toward a customizable space in which it might be easier to compute similarities ($iii.$). It is desirable to select $\psi$ so that it will not be particularly sensitive to small variations of $x$, to promote stability in the matching process, but nothing prevents $\psi$ to be the identity function.[1] There exist several different ways of implementing $\text{sim}$, such as the dot product (scaled by the square root of the key size), the cosine similarity, RBF kernels $\left[\exp\left(-\frac{1}{2\sigma}\|\psi(x) - K_i\|^2\right)\right]_{i=1}^m$, being $\|\cdot\|$ the $L_2$ norm, and others. In our experience, we used the RBF implementation in 2-dim cases, while the the cosine similarity in all the other experiments of this paper. When CMNs belong to the same layer, sharing the exact same input $x$, it is convenient to share $K$ among them, since the key-update operations only depend on $x$. This is not only useful to save memory and computational time, but also to keep learning more stable.

## 2.1 Insights on the Computational Scheme

The foundations of the proposed neuron stands on two widely known topics in the field of machine learning, that are Kernel Methods (Schölkopf et al., 2002) and Attention Models (Vaswani et al., 2017). CMNs also share interesting analogies with Ensemble Methods (Zhou, 2012) and ReLUs (Hara et al., 2015; Xu et al., 2015).

**Kernel Machines.** Let us consider a set of training samples $\{x_1, \ldots, x_h\}$, stored (row-wise) into matrix $X$. The popular Representation Theorem (Schölkopf et al., 2002) in Kernel Methods indicates the form of the optimal solution to a learning problem defined by pointwise constrains (Gnecco et al., 2015) evaluated on the training samples, and regularized in a Reproducing Kernel Hilbert Space induced by the selected kernel function $k(\cdot, \cdot)$, i.e., $f(x, X) = \sum_{i=1}^h \xi_i k(x, x_i)$. This form clearly shows *how* $f$ depends on the given training examples. In turn, since $k(a, b) := \langle \phi(a), \phi(b) \rangle$, being $\phi$ the implicit (usually unknown) feature map of the selected kernel (Schölkopf et al., 2002), we get $f(x, \xi, X) = \sum_{i=1}^h \xi_i \langle \phi(x), \phi(x_i) \rangle = \langle \phi(x), \sum_{i=1}^h \xi_i \phi(x_i) \rangle$, being $\xi$ the vector collecting all the learnable coefficients $\xi_i$'s. In the case of a linear kernel $k(a, b) = \langle a, b \rangle = a'b$, we have that $\phi$ is the identity function, and

$$f(x, \xi, X) = \langle \phi(x), \sum_{i=1}^h \xi_i \phi(x_i') \rangle = \langle x, \sum_{i=1}^h \xi_i x_i \rangle = (X'\xi)'x. \tag{5}$$

Eq. 5 shows that training examples are blended by means of a weighted sum, and then multiplied (dot product) with the input $x$. This is the same form of the computational model used in CMNs, where memories are treated as the training examples in Kernel Methods. In fact, once we replace $X \leftarrow M$ and we set $\xi \leftarrow \alpha(x, K, \delta)$ in Eq. 5, we get back the CMN model of Eq. 2. In Kernel Methods, $\xi$ is learnable, and $X$ is given and fixed. CMNs relax the previous statement in two different manners. First of all, we let the neuron learn to adapt $X$ ($M$) and, second, we let the $\xi$ be the outcome of a function $\alpha$ that depends on $x$.

---

[1]That is what we did in our experiments. There is room for investigating the way $\psi$ could be defined—beyond the scope of this paper. For example, if $x$ is an image/feature map, $\psi(x)$ could be a down-scaling operation.

**Attention & Transformers.** The attentive function $\alpha$ clearly traces the connection with Attention Models, that are frequently described by means of key-value maps, mapping a query to a certain output (Vaswani et al., 2017). The neuron input $x$ represents a query, while memory units in $M$ are the values associated to the set of keys in $K$. Transformers (Vaswani et al., 2017) use (self-)attention to process the information starting from a set of learnable embeddings. Each input embedding is transformed into a new representation, that is further elaborated (usually projected again, normalized, etc.) and passed to the next attention-based encoding block. CMNs borrow the idea of exploiting learnable embeddings blended through an attention mechanism to elaborate the information. Differently from Transformers, CMNs do not use attention to map $x$ onto a novel representation, while they use it to determine *what is the weight vector* $w = \hat{w}(x, K, M)$ that projects $x$ toward the neuron output.

**Ensemble Methods & ReLUs.** Another interesting connection exists between CMNs and Ensemble Methods (Zhou, 2012), where the outputs of a pool of classifiers are combined. Replacing $\hat{w}$ of Eq. 2 with its definition in Eq. 3, we get

$$f(x, K, M) = \alpha(x, K, \delta)' M x = \alpha(x, K, \delta)' \left[ M_i x \right]_{i=1}^{m},$$

that shows how a CMN is virtually a combination (weighted by $\alpha$) of the output of $m$ classic neurons $\left[ M_i x \right]_{i=1}^{m}$, even though the way they are combined is continual learning-oriented due to the way $\alpha$ is defined. Sparse Mixture of Experts can be described following the same notation, where $\alpha$ is implemented by a learnable gating mechanism that selects only $\delta$ experts at once, propagating the gradients towards all of them. The general form of CMNs also intersects the one of classic neurons that exploit the largely known Rectifier Linear Units (ReLUs) (Hara et al., 2015), although in a degenerate case. The output of a ReLU-based neuron is either $w'x$ (when $w'x > 0$) or 0 (otherwise). Let us consider a CMN with $m = 2$ memory units, where $M_1 = w$ and $M_2 = 0$, i.e., a vector of zeros. Let $K_1 = w$ and $K_2 = -w$, and let $\gamma \to \infty$, forcing the softmax function in $\alpha$ to return 1-hot scores only. This implies that $\alpha$ will return $[1, 0]'$ if $w'x > 0$ and $[0, 1]'$ otherwise,[2] that will trigger $M_1 = w$ or $M_2 = 0$, respectively. It is trivial to see that such CMN, without any additional activation function, is equivalent to a a ReLU-based classic neuron. Similar arguments can be used to show the CMN counterpart of leaky-ReLUs (Xu et al., 2015).

# 3 CONTINUAL LEARNING

We consider a stream of data which, at each time instant $t$, yields a data sample $x^{(t)}$, possibly paired with some label information $y^{(t)}$. In principle, gradient-based online continual learning with CMNs is immediately possible once we replace the classic neuron model with the one of Eq. 2. However, in CMNs multiple memory units (up to $\delta$) are blended to generate a weight vector. On one hand, this is important in terms of transfer of information, on the other hand, this might have drawbacks in terms of catastrophic forgetting, since making a gradient step during the latter task will alter all the involved units, and it is hard to say in advance what will be the impact on the previously learned information. Moreover, limiting the computations to the top $\delta$ memory units (Eq. 3) could end up emphasizing just a few common keys, and not using the other ones in $K$.

**Blended Inference, WTA Updates.** In order to overcome the first issue, we follow the most conservative route, updating only the memory unit (and key) associated to the largest attention score, in a Winner-Take-All (WTA) fashion. In this way, CMNs still benefit from the transfer of information at inference stage, while they will focus on specializing only the "winning" memory unit (key) when learning from data. The second issue suggests rethinking the procedure to update $K$, not only in order to avoid degenerate cases (where a single key is used) but also to bias the update dynamics in function of the requirements of continual learning. As a matter of fact, keys can be interpreted as "representatives" that should be located in different regions of the neuron input space, so that they will blend different subsets of memory units, allowing the neuron to eventually behave in a different/specialized manner. For example, when the properties of the streamed data change, a different key is expected to receive large attention, generating a weight vector that mostly depends on a different memory unit. We propose to update the keys in $K$ with an online WTA procedure that is inspired by online K-Means (Zhong, 2005). Consistently with the selected implementation of the similarity function in Eq. 4, if $K_\dagger$ is the winning key for the current $x$, then

$$K_\dagger = K_\dagger + \beta \nabla \text{sim}(\psi(x), K_\dagger), \tag{6}$$

---

[2]The case in which $w'x$ is zero must be specifically handled. As a matter of fact, it is the point which requires a special handling also in ReLUs, to deal with the discontinuity in the first derivative.

where $\beta > 0$ tunes the strength of the update operation. Eq. 6 is a gradient step in the direction that maximizes the similarity between $\psi(x)$ and the winning key, so that the winning key $K_\dagger$ is slightly moved toward the current $\psi(x)$. Attention scores are refreshed according to the updated $K$.

**Avoiding Weak Keys.** In order to favour the development of all the available keys, each CMN computes basic statistical quantities about the way keys are exploited, namely the *absolute usage* $\mu$ and the *absolute age* $\eta$, that are vectors with $m$ components (one-per-key). The former counts how many times each key turned out to be a winning key (usage), while the latter counts how many time steps have passed since the last time a specific key was the winning one (age). The $i$-th key is marked as not used enough if $\mu_i < \tau^\mu$, while it is too old if $\eta_i \geq \tau^\eta$, given two custom positive thresholds $\tau^\mu$ and $\tau^\eta$. We label as *weak* a key that fulfills both the conditions. Vector $\mu$ is initialized with zeros, while the entries in $\eta$ are set to $\tau^\eta$, so that, at the beginning, all keys are weak. Before updating keys with Eq. 6, if there exists at least a weak key and $x$ falls "far away" from $K_\dagger$, i.e., the maximum similarity score returned by sim is smaller than $\tau^\alpha$ (indicating that $x$ is somewhat different from all the entries of $K$), the CMN replaces the weak key with $x$ itself (actually $\psi(x)$), zeroing its usage counter.[3] Moreover, instead of randomly re-initializing the memory unit associated to the replaced key, the CMN initializes it to $M_\dagger$, to favour a warm-start of the memory values. Algorithm 1 summarizes continual learning in a CMN (see Appendix A for the case of mini-batches).

---

**Algorithm 1** Learning with a Continual Memory Neuron: WTA updates of memory units and keys.

---

**Require:** Stream $\mathcal{S}$; generic loss function $\mathrm{loss}(\ldots)$; learning rates $\rho, \beta$; $K \leftarrow$ rand, $M \leftarrow$ rand, $\mu \leftarrow$ 0's, $\eta \leftarrow \tau^\eta$'s.

  **while** $true$ **do**
    $x \leftarrow \mathrm{next\_neuron\_input}(\mathcal{S})$
    $a, s \leftarrow \alpha(x, K, \delta)$              ▷ Attn. scores (Eq. 4) and similarities
    $\dagger \leftarrow \arg\max_{j \in \{1,\ldots,m\}}\{a_j\}$         ▷ Index of the winning key
    $K, M, \dagger, \mu \leftarrow$
        $\mathrm{SCRAMBLE}(x, K, M, s, \dagger, \mu, \eta)$     ▷ Possibly replace a key
    $K_\dagger \leftarrow K_\dagger + \beta \nabla \mathrm{sim}(\psi(x), K_\dagger)$     ▷ Upd. winning key, Eq. 6
    $a \leftarrow \mathrm{REFRESH}(x, s, K_\dagger, \dagger)$          ▷ Refresh $a$ (dummy)
    $\mu_\dagger \leftarrow \mu_\dagger + 1, \ \ \eta_\dagger \leftarrow 0$     ▷ Increase $K_\dagger$ usage, reset its age
    $\eta = \eta + 1$                          ▷ Increase *all* ages
    $w \leftarrow M'a$                     ▷ Generate weights, Eq. 3
    $o \leftarrow w'x$                  ▷ Compute neuron output, Eq. 2
    $M_\dagger \leftarrow M_\dagger - \rho \nabla_{M_\dagger} \mathrm{loss}(o)$     ▷ Upd. winning memory unit
  **end while**

---

**function** $\mathrm{SCRAMBLE}(x, K, M, s, \dagger, \mu, \eta)$
  $\mathcal{W} \leftarrow \{z: \mu_z < \tau^\mu \wedge \eta_z \geq \tau^\eta\}$      ▷ Indices of weak keys
  **if** $s_\dagger < \tau^\alpha \wedge \mathcal{W} \neq \emptyset$ **then**     ▷ If the current match is loose
    $j \leftarrow \arg\max_{z \in \mathcal{W}}\{\eta_z\}$     ▷ The weakest key is the oldest
    $K_j \leftarrow \psi(x)$                ▷ Replace weakest key
    $M_j \leftarrow M_\dagger$      ▷ Warm-start for replaced memory unit
    $\mu_j \leftarrow 0$                   ▷ Reset usage counter
    $\dagger \leftarrow j$                    ▷ Update winning key index
  **end if**
  **return** $K, M, \dagger, \mu$
**end function**

---

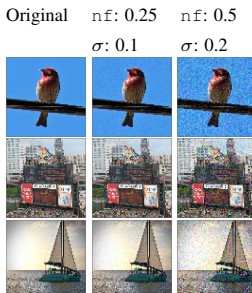

Original   nf: 0.25   nf: 0.5
               $\sigma$: 0.1     $\sigma$: 0.2

**Figure 2:** Samples from the NS-IMAGENET dataset. Samples are perturbed to create different distributions by adding Gaussian noise with noise factor `nf` and standard deviation $\sigma$.

## 4 EXPERIMENTS

**Setup.** CMNs do not introduce any constraints on the specific type of continual learning problem to be faced, differently from what it is indeed common in the existing literature (task incremental, class incremental, domain incremental, etc. (van de Ven et al., 2022; Delange et al., 2021)). For experimental purposes, we consider a setting that is structured to make it more controllable, but *without using any of its properties for learning purposes*. Data is continuously streamed from a source $\mathcal{S}$, that, at each time instant $t$, yields a single example $x^{(t)}$ and its class label information $y^{(t)}$, learning in a single-pass online manner. The lifetime $[1, T]$ of the learning agent is partitioned into $N$ non-overlapping ranges, $[1, T] = \cup_{j=1}^N I_j$, with $I_j \cap I_r = \emptyset, \forall j, r$, featuring data sampled from different distributions. We simulate three types of experiences, namely CI (*class incremental*), CDI (*class and domain incremental*), CDID (*class and domain incremental with dependent sampling*), exploiting data belonging to $c$ classes, as we describe in the following. In CI, each time interval is about data from one class, $N = c$. Data is sampled in an independent manner within each interval.

---

[3]There might be multiple weak keys. We select the oldest one.

CDI provides one class after the other for $c$ contiguous intervals, then the class distributions changes and the process is repeated multiple times over the same classes. We consider $q$ different distributions per class, thus $N = cq$. CDID is the same as CDI, but data is sampled in a *dependent* manner within each interval. In all the experiments of this paper, we selected the optimal values of the hyper-parameters by maximizing the average accuracy on a held-out validation set after having processed $N/2$ intervals–coherently with recent work (Cai et al., 2021) and differently from approaches that identify the best models on test data (Lopez-Paz & Ranzato, 2017; De Lange & Tuytelaars, 2021).

**Data, Metrics.** We sampled 2D data, referred to as MODES, belonging to $c = 4$ mutually exclusive categories, where each class involves data distributed as a bi-modal distribution (the two modes are jointly used in CI, while separately sampled to simulate the CDI, $q = 2$), while the CDID setting is implemented by sorting data samples for ascending values of their first coordinate. The case in which the two Gaussian modes are replaced with two arc-shaped distributions is also considered, named MOONS (Fig. 4-left). Additionally, we performed experiments on two standard CL benchmarks, hereinafter referred to as MNIST-CI and CIFAR10-CI, that follow the CI setting. Both the experiences were characterized by $c = 10$ and $q = 1$. We also considered a real-world dataset proposed for continual online learning in (Mai et al., 2022b), NonStationary-MiniImageNet, indicated with NS-IMAGENET, designed for the challenging CDI scenario. It is composed of images belonging to $c = 100$ classes, at the resolution of $84 \times 84$. We considered $q = 3$ domains, sampling data from the original distribution plus two perturbed distributions adding Gaussian noise (see Fig. 2). In total, we obtain 180K images, 20% of which are used for testing and 10% for validating. Coherently with the other experiences of this paper, we used validation data only from the first $N/2$ intervals (recall $N = cq$). For all experiments, performance is measured using common metrics, as defined in (Mai et al., 2022b): average accuracy at time $T$, average forgetting at time $T$, as well as forward transfer (backward transfer was not significant in our comparisons). See Appendix B.

**Compared Models.** We compared a CMN net equipped with a layer of CMNs with architectures that store samples in a memory buffer (one hidden layer with $h$ hidden neurons, hyperbolic tangent activation). Buffers are either used for continuous replay purposes, such as Experience Replay (ER) (Rolnick et al., 2019), using Random Sampling (RND) or the more advanced Reservoir (RES) Sampling (Chrysakis & Moens, 2020), or to set up the GDUMB approach (Prabhu et al., 2020), that is focused on retraining the network from the buffer data only, keeping it balanced with respect to the class labels. We also compared to state-of-the-art continual learning approaches such as MIR (Aljundi et al., 2019a), that features specifically designed retrieval operations on the buffer, and A-GEM (Chaudhry et al., 2019), with gradient-based memories, ASER (Shim et al., 2021), BIC-inspired Knowledge Distillation (Mai et al., 2022a), DER and DER++ (Buzzega et al., 2020). These methods have been proven to represent the SOTA in online continual learning without explicit task boundaries (Mai et al., 2022a; Wang et al., 2023). We also considered ENSEMBLE models (Zhou, 2012), where the outputs of 10 networks are combined either by averaging them or learning a gating function (Mixture of Experts (MOE) (Shazeer et al., 2017)), and VANILLA networks with classic neurons.

In Appendix C we formally analyze and compare the memory and computational burden of CMN-networks vs. classic-nets, to setup a fair comparison (Zhou et al., 2023). In MODES and MOONS we set $m = 8$ memory units (the max number of intervals $N$), and we equipped competitors with a buffer of size 8, $h \in \{5, 25\}$. Notice that even when replaying only a sample per step, the computational burden of the competitors is always larger than the CMN-net. In MNIST-CI, CIFAR10-CI, and NS-IMAGENET we considered $m \in \{10, 25, 50, 100\}$, while competitors leverage a $10c$-exemplars buffer (i.e., 100, 100, and 1000 exemplars, respectively) sampled with a batch size of 10 samples, following the experience of Mai et al. (2022b), and an architecture with $h \in \{50, 100\}$. In NS-IMAGENET, a ResNet50 backbone processed the images, extracting representations of size $u = 512$. See Appendix D for a detailed description of the grid of values for each hyper-parameter and the resulting optimal values. The code of CMNs can be found at `http://see-attached-code`.

**Results.** We first compare the average test accuracy at time $T$ on the 2D datasets, MODES and MOONS in Fig. 3. Our CMN net, thanks to its pool of weights, is able to effectively separate the data, achieving an average accuracy that doubles the performance of most competitors on the MODES dataset. A similar behavior is observable for the MOONS dataset, although the gap between CMN and other replay memory is thinner. It can be noted how the vanilla networks are not able to capture the non-stationary nature of the data distributions, while some competitors equipped with memory buffers can almost double its performance. CMN is able to overcome all of them both in the CDI and CI settings, confirming its capability of driving the memorization of the properties of the data distributions in

**Table 1:** MNIST-CI, CIFAR10-CI, NS-IMAGENET. Performance on the test data (**1st-best**, 2nd/3rd-best).

| MODEL | MNIST-CI | | | CIFAR10-CI | | | NS-IMAGENET | | |
|---|---|---|---|---|---|---|---|---|---|
| | ACC. ↑ | FORGET ↓ | FWD. T. ↑ | ACC. ↑ | FORGET ↓ | FWD. T. ↑ | ACC. ↑ | FORGET ↓ | FWD. T. ↑ |
| VANILLA | 31.5 ± 1.5 | 23.2 ± 8.1 | 0.4 ± 0.7 | 19.8 ± 0.8 | 60.1 ± 1.3 | 1.4 ± 0.2 | 6.8 ± 0.6 | 13.2 ± 0.6 | 1.6 ± 0.1 |
| AGEM | 35.5 ± 1.9 | 15.8 ± 9.2 | 0.4 ± 0.7 | 20.5 ± 0.7 | 58.0 ± 1.4 | 1.4 ± 0.2 | 6.4 ± 0.3 | 93.9 ± 0.3 | 2.0 ± 0.1 |
| ASER | 70.2 ± 0.4 | 30.7 ± 0.4 | 0.0 ± 0.0 | 25.2 ± 1.4 | 40.0 ± 3.4 | 1.0 ± 0.3 | 54.0 ± 0.2 | 32.3 ± 0.4 | 21.0 ± 0.3 |
| BIC-KD | 46.6 ± 0.9 | **3.9 ± 0.6** | 0.0 ± 0.0 | 24.7 ± 0.6 | **4.3 ± 0.1** | 2.2 ± 0.4 | 13.0 ± 0.6 | **1.1 ± 0.1** | 4.6 ± 0.3 |
| DER | 77.6 ± 0.8 | 23.8 ± 0.9 | 0.0 ± 0.0 | 26.1 ± 1.0 | 58.3 ± 2.1 | 2.1 ± 0.3 | 46.5 ± 4.9 | 53.5 ± 5.0 | 15.6 ± 2.1 |
| DER++ | 78.2 ± 1.2 | 23.0 ± 1.4 | 0.0 ± 0.0 | 26.9 ± 0.9 | 51.9 ± 2.2 | **2.3 ± 0.3** | 55.9 ± 1.0 | 43.4 ± 1.1 | **25.0 ± 0.4** |
| ER-RANDOM | 21.0 ± 5.6 | 73.3 ± 8.3 | 0.1 ± 0.1 | 17.3 ± 1.4 | 79.4 ± 3.9 | 1.0 ± 0.4 | 22.4 ± 0.1 | 76.8 ± 0.3 | 9.0 ± 0.1 |
| ER-RESERVOIR | 70.3 ± 0.6 | 32.0 ± 0.7 | 0.0 ± 0.0 | 24.0 ± 1.3 | 46.4 ± 0.4 | 0.9 ± 0.1 | 49.9 ± 0.3 | 49.4 ± 0.3 | 21.5 ± 0.0 |
| ENSEMBLE | 49.8 ± 9.3 | 40.8 ± 9.6 | 0.0 ± 0.0 | 20.5 ± 0.4 | 63.9 ± 0.4 | 1.0 ± 0.2 | 22.9 ± 2.2 | 35.2 ± 1.5 | 4.1 ± 0.4 |
| GDUMB | 14.3 ± 1.7 | 43.3 ± 7.2 | **0.9 ± 0.7** | 18.1 ± 1.0 | 28.2 ± 2.8 | 1.5 ± 0.5 | 6.0 ± 0.6 | 61.4 ± 1.3 | 4.7 ± 0.0 |
| MIR | 70.9 ± 0.3 | 31.4 ± 0.4 | 0.0 ± 0.0 | 23.8 ± 1.5 | 46.6 ± 0.4 | 0.9 ± 0.1 | 53.7 ± 0.3 | 45.8 ± 0.3 | 24.4 ± 0.2 |
| MOE | 23.2 ± 10.1 | 64.4 ± 15.2 | 0.7 ± 0.3 | 18.4 ± 0.3 | 59.6 ± 2.9 | 1.4 ± 0.5 | 1.7 ± 0.7 | 86.3 ± 1.7 | 1.2 ± 0.0 |
| CMN (Ours) | **78.8 ± 0.2** | 13.9 ± 0.2 | 0.5 ± 0.4 | **27.9 ± 0.3** | 26.3 ± 0.4 | 1.8 ± 0.2 | **57.8 ± 0.1** | 24.3 ± 0.1 | 24.2 ± 0.1 |

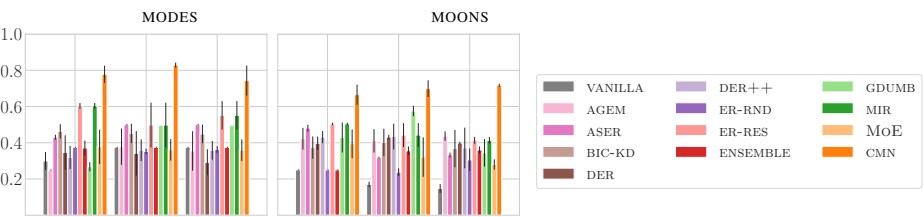

**Figure 3:** MODES and MOONS data, test accuracy and std in the three setting we analyzed (CI, CDI, CDID).

an effective manner. For completeness, in Appendix F we show that even when selecting the best hyper-parameters according to the test set, the CMN net still overcomes the competitors. Switching to the more demanding datasets (MNIST-CI, CIFAR10-CI, NS-IMAGENET), results in Tab. 1 show that we achieve the highest accuracy across all competitors. Interestingly, in the case of the NS-IMAGENET and across all the models, only CMN, MIR, DER, ASER, and ER-RESERVOIR are able to provide appreciable forward transfer capabilities, confirming that the weights learned by our method can indeed be shared across distributions and effectively used throughout training. More importantly, CMN is among the few methods that do not dramatically suffer from forgetting issues. Only VANILLA yields less forgetting, paired with lower accuracy (coherently with (Mai et al., 2022b)).

**Insights.** Since MODES and MOONS are two-dimensional datasets, we can inspect what the network is learning. In Fig. 4-left we show how the decision boundaries are shifting during training. Interestingly, the CMN model is adapting portions of the space to different distributions, incrementally as they are observed. At the same time, the VANILLA counterpart forgets previously learned boundaries in favor of the last distributions that are shown to the model. We also plot the keys of the continual memory neurons in the 2D-plane. As expected, keys shift during training to effectively cover the modes of the distributions, ignoring empty regions. In fact, we can trace which keys are used throughout the learning process. In Fig. 4-right we show a histogram of winning key activations for each interval in the lifespan of the agent, i.e. when data of a certain distribution is presented to the model. Keys are mostly non-overlapping across distributions (MOONS data are less spread over the input space), supporting the hypothesis of parameter isolation that we aimed to achieve with CMNs.

**Ablations.** We perform an ablation study to quantify the importance of the components of the proposed approach, focusing on the strategies presented in Section 2: WTA updates and the key update strategy of Eq. 6. In Fig. 5 (a-b) we compare our proposed approach against modified versions of CMN. We denote with $a_M$ a global update involving memories corresponding to *a*ll the top-$\delta$ keys instead of the best-scoring one (as in the WTA approach). Similarly, $g_K$ indicates a *g*radient-based update of the keys instead of using the proposed update strategy. It can be seen how switching off WTA updates on memories ($a_M$) has the effect of favoring forgetting since multiple sets of memories are updated at the same time, leading to lower accuracies. Gradient-based updates of keys instead yield an even worse drop in accuracy consistently across all datasets with a more pronounced amount of forgetting. We attribute this result to a less predictable behavior of keys when using gradient-based updates. In fact, the amount of update for a key is modulated by the quality of the predictions since gradients scale with the loss. This can lead to unwanted behaviors with respect to Eq 6, such as too big updates or no update at all, depending on the fitting quality of each individual sample. As one would expect, combining both $g_K$ and $a_M$ does not help. In Figure 5 (c-d) we investigated the role

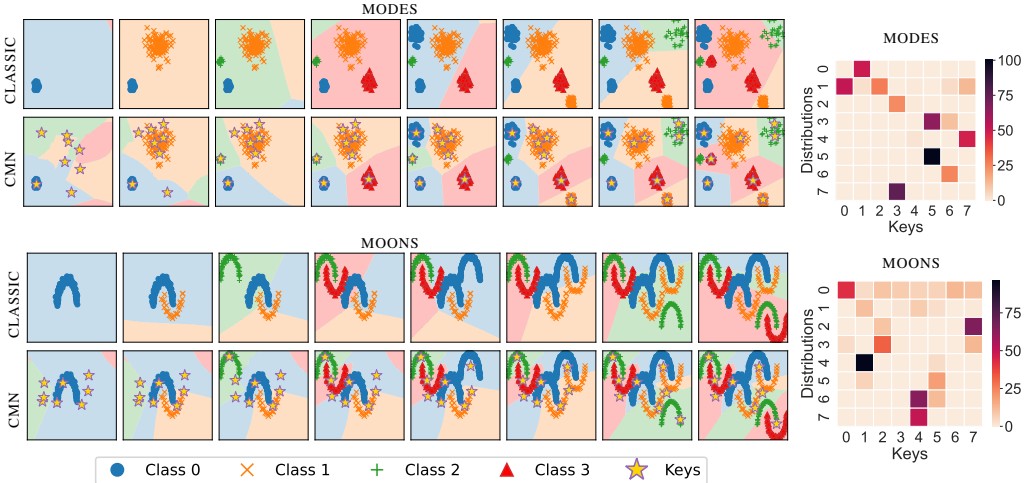

**Figure 4:** Left: Decision boundaries over time (left-to-right) with different learning settings, in MODES and MOONS. Samples are provided one-by-one and each picture is relative to the timestep where a new distribution is completely processed (i.e., after having processed the last sample in each distribution). We compare CMN with classic neurons. Right: Data distributions of MODES and MOONS along with key activations during training in a CDI setting. Each interval in the lifespan of the agent (corresponding to a different data distribution) activates different keys favoring parameter isolation.

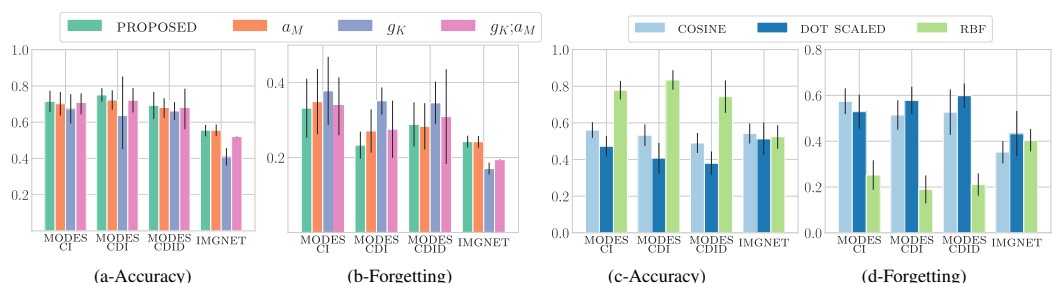

**Figure 5:** Ablation studies. (a-b): proposed approach against modified versions of CMN. $a_M$ is a global update that involves the *M*emories corresponding to *a*ll the top-$\delta$ keys instead of the winning one only; $g_K$ indicates a *g*radient-based update of the *K*eys instead of the proposed update strategy. (c-d): Similarity function. In both the cases, we report the average accuracy at $T$ (a-c, **higher** is better) and forgetting at $T$ (b-d, **lower** is better).

of the similarity function sim (Eq. 4). In 2-dim datasets, the RBF kernel implementation is the one that provides the best performance, as expected. In real-world experiences, by leveraging cosine similarity the keys can better disentangle the high-dimensional data representation.

## 5    CONCLUSIONS AND FUTURE WORK

We proposed a novel model of neuron for continual learning purposes, named Continual Memory Neuron (CMN). CMNs are memory-equipped neurons that can generate their own weights based on the current input, aiming for plasticity but less subject to catastrophic forgetting issues. We evaluated CMNs in continual learning benchmarks, showing that CMN-based nets overcomes classic-neuron-based nets, even when they exploit additional memory buffers and replays. These results open to a novel research perspective, since, to our best knowledge, focusing on a novel neuron model is different to what is commonly done in related works. In principle, CMNs can be plugged into any neural architecture. Their interactions in different classes of deep networks and the way they behave with other learning approaches will be the subject of future work.

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

# A BATCHED COMPUTATIONS

In Algorithm 2 we report the same operations of Algorithm 1 when a mini-batch of size $b$ is provided by the stream at each time instant. The case of batched computations is pretty straightforward, with a major difference in the key scrambling routine. As a matter of fact, scrambling would require to serialize the processing of the element in the mini-batch, that is not a desirable feature. For this reason, we restrict to 1 the number of weak keys that can be potentially replaced for each mini-batch, selecting the one associated to the batch element that is returning the smallest similarity score with respect to $K$.

---

**Algorithm 2** Learning with a Continual Memory Neuron when a batch of $b$ samples is provided at each time instant by the stream. Notice that $X$ is the mini-batch matrix ($b \times d$), function $\alpha$ is intended to compute attention scores for each element of the batch, $\dagger$ and $o$ are now arrays of length $b$. Scrambling involves up to 1 key for each mini-batch (function SCRAMBLEONE).

---

**Require:** Stream $\mathcal{S}$; generic loss function loss$(\ldots)$, learning rates $\rho, \beta$; $K \leftarrow$ rand, $M \leftarrow$ rand, $\mu \leftarrow 0$'s, $\eta \leftarrow \tau^\eta$'s.

  **while** *true* **do**

    $X \leftarrow$ next_neuron_inputs$(\mathcal{S})$                                            ▷ $X$ is a $b \times d$ matrix

    $A, S \leftarrow \alpha(X, K, \delta)$                                     ▷ $A$ and $S$ are $b \times m$ matrices

    $\dagger_h \leftarrow \arg\max_{j \in \{1,\ldots,m\}}\{A_{hj}\}, \ h = 1, \ldots, b$

                                                    ▷ Indices of the winning keys

    $K, M, \dagger, \mu \leftarrow$ SCRAMBLEONE$(X, K, M, S, \dagger, \mu, \eta)$

                                          ▷ Possibly replace a weak key

    $K_{\dagger_h} \leftarrow K_{\dagger_h} + \beta \nabla \text{sim}(\psi(X_h), K_{\dagger_h}), \ h = 1, \ldots, b,$

                                            ▷ Upd. winning keys, Eq. 6

    $A \leftarrow \alpha(X, K, \delta)$                                 ▷ Refresh attention (from scratch)

    $\mu_{\dagger_h} \leftarrow \mu_{\dagger_h} + 1, \quad \eta_{\dagger_h} \leftarrow 0, \quad h = 1, \ldots, b$

                                     ▷ Increase winning keys usages, reset ages

    $\eta = \eta + b$                                              ▷ Increase *all* ages

    $W \leftarrow AM$                             ▷ Generate weights, Eq. 3, $W$ is a $b \times d$ matrix

    $o_h \leftarrow W_h X_h', \quad h = 1, \ldots, b$                     ▷ Compute output, Eq. 2

    $M_\dagger \leftarrow M_\dagger - \rho \nabla_{M_\dagger} \text{loss}(o)$            ▷ Upd. winning memory unit

  **end while**

---

  **function** SCRAMBLEONE$(X, K, M, S, \dagger, \mu, \eta)$

    $\mathcal{W} \leftarrow \{z \colon \mu_z < \tau^\mu \wedge \eta_z \geq \tau^\eta\}$                       ▷ Set of weak keys (if any)

    $k \leftarrow \arg\min_{h \in \{1,\ldots,b\}}\{S_{h\dagger_h}\}$

                           ▷ Idx of the sample with lowest similarity to its winning key

    **if** $S_{k\dagger_k} < \tau^\alpha \wedge \mathcal{W} \neq \emptyset$ **then**            ▷ If the current match is loose

      $j \leftarrow \arg\max_{z \in \mathcal{W}}\{\eta_z\}$                      ▷ The weakest key is the oldest

      $K_j \leftarrow \psi(X_k)$                   ▷ Replace weakest key with data sample

      $M_j \leftarrow M_\dagger$                   ▷ Warm-start for replaced memory unit

      $\mu_j \leftarrow 0$                               ▷ Reset usage counter

      $\dagger_k \leftarrow j$                            ▷ Update winning key index

    **end if**

  **return** $K, M, \dagger, \mu$

  **end function**

---

# B METRICS

Following the notation of Section 4, we are given a data stream $\mathcal{S}$ partitioned into non-overlapping time intervals, and we indicate with $t_j$ the last time instant of the $j$-th interval $I_j$, with $t_N = T$. In the following description, we assume class indices to be ordered with respect to the time in which they become available, to keep the notation simple. We indicate with $p(x|I_i)$ the data distribution in the $i$-th interval, with $\theta_j$ is the model developed up to $t_j$ (being it a CMN-based network or the network of another competitor), while $D_i$ is an held out test set with data sampled from $p(x|I_i)$. Then, we indicate with

$$\text{acc}_i^{\theta_j} = \text{accuracy}(\mathcal{D}_i, \theta_j)$$

the accuracy on data sampled from $p(x|I_i)$ computed using the model parameters at $t_j$, i.e., $\theta_j$. We collect the following matrix of accuracies during the learning procedure,

$$
\begin{bmatrix}
\begin{array}{c|cccccc}
\text{Model/Test Data} & \mathcal{D}_1 & \mathcal{D}_2 & \dots & \mathcal{D}_j & \dots & \mathcal{D}_N \\
\hline
\theta_1 & \text{acc}_1^{\theta_1} & \text{acc}_2^{\theta_1} & \dots & \text{acc}_h^{\theta_1} & \dots & \text{acc}_N^{\theta_1} \\
\theta_2 & \text{acc}_1^{\theta_2} & \text{acc}_2^{\theta_2} & \dots & \text{acc}_j^{\theta_2} & \dots & \text{acc}_N^{\theta_2} \\
\dots & \dots & \dots & \dots & \dots & \dots & \dots \\
\theta_j & \text{acc}_1^{\theta_j} & \text{acc}_2^{\theta_j} & \dots & \text{acc}_j^{\theta_j} & \dots & \text{acc}_N^{\theta_j} \\
\dots & \dots & \dots & \dots & \dots & \dots & \dots \\
\theta_N & \text{acc}_1^{\theta_N} & \text{acc}_2^{\theta_N} & \dots & \text{acc}_j^{\theta_N} & \dots & \text{acc}_N^{\theta_N}
\end{array}
\end{bmatrix}
$$

that we indicate as continual confusion matrix (CCM, being $\text{CCM}_j$ the matrix up to $t_j$), and we exploit it to compute the following measures. Notice that it is a square matrix.

- The average accuracy at $t_z$ is defined as the average of the $z$-th row of the CCM, up to the $z$-th column (included),

$$
\text{avg\_accuracy}(\text{CCM}_z) = \frac{1}{z} \sum_{i=1}^{z} \text{acc}_i^{\theta_z},
$$

  and we commonly measure the average accuracy (Acc of Section 4) at the end of training, $t_z = t_N$.

- The average forgetting at $t_z$ can be defined as

$$
\text{avg\_forgetting}(\text{CCM}_z) = \frac{1}{z-1} \sum_{i=1}^{z-1} \left( \text{acc}_i^{\star} - \text{acc}_i^{\theta_z} \right),
$$

  where $\text{acc}_i^{\star}$ is the best accuracy obtained on data $\mathcal{D}_i$ so far, i.e., $\max_{\theta_k \in \{\theta_1,\dots,\theta_{z-1}\}} \text{acc}_i^{\theta_k}$ (maximum of the $i$-th column up to row $z-1$). We commonly measure the average forgetting (Forg of Section 4) at the end of training, $t_z = t_N$.

- Forward transfer measures how learning at checkpoint $t_z$ (positively) influences predictions on data introduced in future intervals,

$$
\text{forward}(\text{CCM}_z) = \frac{2}{z(z-1)} \sum_{i=1}^{z-1} \sum_{j=i+1}^{z} \text{acc}_i^{\theta_j},
$$

  i.e., it is the average of the upper-triangular portion of the CCM (excluding the diagonal). Similarly to the previous cases, we commonly measure it at $t_z = t_N$, yielding Fwd of Section 4.

## C  Computational Cost

A classic neuron model in a neural network requires $u$ products to compute the output score, being $u$ the size of the input, Eq. 1. We compare this cost in terms of the operations performed by CMNs, still using products as a reference. Computing the output of a CMN involves three main operations: (1) evaluating $\psi(x)$, whose cost is $C(\psi)$, that transforms the neuron input in a customizable manner, being $\tilde{u}$ the size of the $\psi$-output; (2) computing the attention scores by $\alpha$, Eq. 4, with cost $m\tilde{u}$ plus the cost of the $\text{softmax}^\delta$ operation, that is $\delta$; (3) blending memories, $\delta u$ products, due to the sparsity of the attention scores; (4) computing the usual output function as in a classic neuron, $u$ products. In total, we have $C(\psi) + m\tilde{u} + \delta + \delta u + u$. The cost of a layer of $n$ classic neurons trivially becomes $un$, while the cost of a layer of CMNs that share the same $K$ is

$$
C(\psi) + m\tilde{u} + \delta + \delta un + un, \tag{7}
$$

where only the last two terms depends on $n$, since the first three ones are about operations that are performed only once, being $K$ shared. In order to reasonable relate the cost of classic and CM neurons, some basic considerations must be introduced. First of all, the cost $C(\psi)$ is expected to be way smaller than the cost of the whole layer. For example, when $\psi$ is just limited to the

$L_2$ normalization of $x$. Moreover, depending on the considered problem, there could be room for designing $\psi$ such that $\tilde{u}$ is smaller than $u$. Of course, this does not always hold. It is reasonable to assume the term $\delta$ in Eq. 7 to be way smaller than the other ones (being it a strong sparsity index, always $< m$), thus we discard it. As a result, we can compute the ratio $R$ between the cost of a CMN layer and the corresponding classic layer,

$$R_C \;=\; \frac{m\tilde{u} + (\delta + 1)un}{un} = \frac{m\tilde{u}}{nu} + \delta + 1. \tag{8}$$

In case of multi-layer nets, with $\ell$ layers, we have

$$R_C \;=\; \frac{\sum_{i=1}^{\ell} m_{(i)}\tilde{u}_{(i)} + (\delta_{(i)} + 1)d_{(i)}n_{(i)}}{\sum_{i=1}^{\ell} d_{(i)}n_{(i)}}, \tag{9}$$

being $i$ the layer index. In terms of memory consumption, a layer of $n$ CMNs stores matrix $K$ and $n$ matrices of memory units $(M)$, that is a total of $m\tilde{u} + mun$ floating point numbers, while in a classic layer only the weight matrix is stored ($un$ floating point values). The ratio $R_M$ for $\ell$ layer is then,

$$R_M \;=\; \frac{\sum_{i=1}^{\ell} m_{(i)}\tilde{u}_{(i)} + m_{(i)}u_{(i)}n_{(i)}}{u_{(i)}n_{(i)}} \tag{10}$$

while the additional memory usage introduced by $\ell$ CMN layers is

$$U_M = m_{(i)}\tilde{u}_{(i)} + (m_{(i)} - 1)u_{(i)}n_{(i)}. \tag{11}$$

A candidate way to compare CMN-based net with models that replay data from memory buffers, is to use the exact same network architecture, using classic neurons in place of CMNs. Then, we allow replay-based methods to sample $R_C - 1$ examples from the buffer at each time step. In fact, these buffer-based models make a prediction on a mini-batch of buffer data in addition to the currently streamed sample, according to the continual online learning setting experimented in this paper. Of course, when comparing with models that have more layers that the CMN-net, it is harder to keep a perfect balance in term of computational cost, so we allowed competitors to have a cost that is slightly larger than the one of the CMN-net, making the comparison more challenging. Moreover, we recall that the replay-based methods learn by exploiting the label-related information they store on the replay-buffer, while no-label-information is stored by the CMN-net (this the comparison becomes unfair when using very large replay buffers).

## D  HYPER-PARAMETERS

We evaluated multiple combinations of values for the main hyper-parameters of CMNs and competitors, that we summarize in the following, in addition to the already described parameter values of the main paper. In the case of CMN-based nets, in MODES and MOONS, we selected $m = 8$ memory units with $\delta = 2$, while we tested $\beta \in \{10^{-4}, 10^{-3}, 10^{-2}, 1\}$, $\tau_\mu \in \{50, 200\}$, $\tau_\eta \in \{50, 200\}$, $\tau_\alpha \in \{0.85, 0.95\}$, $\gamma \in \{1, 5, 25\}$. In NS-IMAGENET we considered $m \in \{10, 25, 50, 100\}$, $\delta \in \{2, 5\}$, $\beta \in \{10^{-3}, 10^{-2}\}$, $\tau_\mu \in \{50, 500, 5000\}$, $\tau_\eta \in \{50, 500, 5000\}$, $\tau_\alpha \in \{0.7, 0.85, 0.95\}$, $\gamma \in \{1, 5, 25\}$. The hidden layer size has been evaluated in $h \in \{5, 25\}$ for 2D data, while in $h \in \{50, 100\}$ for NS-IMAGENET. In all the models, we considered a learning rate $\rho \in \{10^{-4}, 10^{-3}, 10^{-2}, 1\}$, and trained with fixed-step-size gradient descent. We also evaluated the case of Adam, which yielded lower results on average. Indeed, adopting optimizers with memory such as Adam may be tricky: at every step, the model might select a different set of weights to be updated, making the statistics of the optimizer invalid. We leave the investigation about the effect of such optimizers for future work, restricting our analysis to memoryless optimizers, which do not suffer from this issue. We also considered a weight decay factor DECAY for the optimizers $\in \{10^{-4}, 10^{-3}, 0\}$. We trained GDumb for 10 epochs on the buffer data. Other minor internal parameters of the competitors were set to the values suggested in the respective papers. The results reported in the main paper are averaged over three runs with different seeds in $\{1234, 123456, 12346578\}$. For all the experiments in this work we used PyTorch, running on a Linux machine–NVIDIA GeForce RTX 3090 GPU (24 GB).

### D.1  OPTIMAL HYPER-PARAMETERS

We report in Table 2 the best selected hyperparameters for the CMN model in all the considered datasets and settings described in the main paper.

**Table 2:** Optimal parameters. The best selected hyperparameters for the proposed CMN model, drawn from the grids described in the text, for the datasets. See the code for further details.

| Parameters | MODES | | | MOONS | | | NS-IMAGENET |
|---|---|---|---|---|---|---|---|
| | CI | CDI | CDID | CI | CDI | CDID | - |
| $\delta$ | 2 | 2 | 2 | 2 | 2 | 2 | 5 |
| $\beta$ | $10^{-2}$ | $10^{-2}$ | $10^{-2}$ | $10^{-2}$ | $10^{-2}$ | $10^{-1}$ | $10^{-3}$ |
| $\rho$ | $10^{-2}$ | $10^{-1}$ | $10^{-2}$ | $10^{-1}$ | $10^{-2}$ | $10^{-2}$ | $10^{-4}$ |
| $\gamma$ | 25 | 25 | 25 | 25 | 25 | 5 | 1 |
| $m$ | 8 | 8 | 8 | 8 | 8 | 8 | 100 |
| $\tau_\alpha$ | 0.95 | 0.95 | 0.95 | 0.95 | 0.95 | 0.85 | 0.7 |
| $\tau_\eta$ | 50 | 50 | 50 | 50 | 200 | 200 | 5000 |
| $\tau_\mu$ | 50 | 50 | 50 | 50 | 50 | 50 | 500 |
| DECAY | 0. | 0. | 0. | 0. | 0. | $10^{-3}$ | $10^{-3}$ |

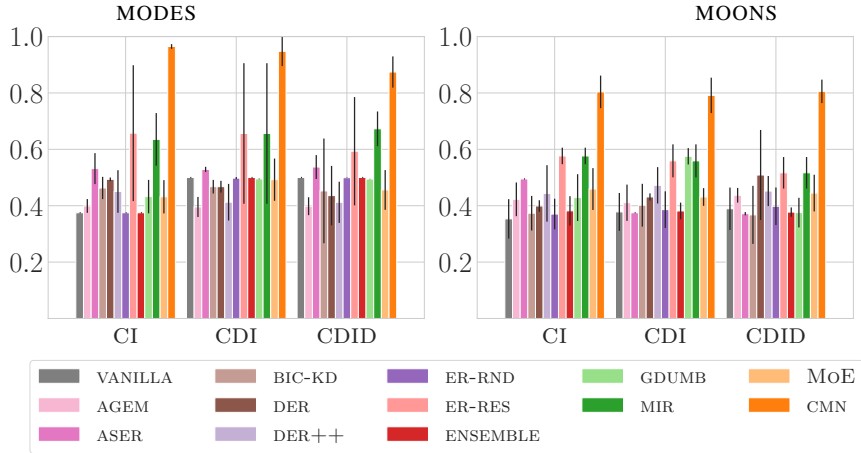

**Figure 6:** MODES and MOONS data, **best** test accuracy (reference only) and std in the three setting we analyzed (CI, CDI, CDID).

## E  NETWORKS OF CMNS

We report in Figure 7 a more detailed view of the process of input projection, memory blending and WTA update occurring in each CMN. We now briefly discuss the specific case of convolutional layers and on networks with multiple stacked CMN-based layers.

In a convolutional layer, each spatial coordinate is associated with a neuron[4] whose input is differently shifted with respect to every other neuron of the layer. At a first glance, this makes less obvious how to share keys (if need) among the neurons of the layer. However, neurons are still expected to share the same weights/filter, thus the key-attention scores should be the same for all of them, in order to coherently blend memories and generate the same filter. In this case, it is convenient to go back to the standard definition of convolution operation, where the whole input map of the layer is one of its operands, thus $\psi(x)$ is a projection of the whole input of the layer, and it is the same for all the neurons. As a result, the attention scores are computed only once per layer, independently of the resolution of the input map.

Layers of CMNs can be stacked into multi-layer networks, as usual. However, while the input of the first layer is not affected by CMN dynamics, the input of any other layer comes from the CMNs of the layer below. The proposed WTA key update scheme is not gradient-based, so that we also avoid gradients to propagate through the key-matching process, i.e., we consider $\hat{w}(x, K, M)$ of Eq. 2 to not depend on $x$ for gradient computation purposes. As a result, gradients flow from the

---

[4]We consider a single filter/output-feature-map in this description, for simplicity. For the same reason, and without any loss of generality, we describe infinitely supported filters.

output layer down to the output of the layer below, as usual, and not through $\hat{w}$. Another intuition that we followed is that CMNs belonging to the lower layers in a deep convolutional network should have a smaller number of memory units than CMNs on the top layers, since lower-level features are likely to be more shared across inputs than the higher-level ones, where the semantics emerge (e.g., edge-like filters in the lowest levels vs. objects/object-parts related filters in the highest ones). Of course, in non-convolutional nets this is harder to say in advance, but we still follow the same intuition motivated by the need of reducing the variability in the outputs of the lower layers, to favor stability in the learning process of the upper layers. Investigating the interaction among layer requires specific studies that go beyond the scope of this paper. Indeed, we focused on networks in which the input of the CMNs was not affected by the learning dynamics, both when using backbones with several layers or when directly classifying data in their original representations. Each CMN virtually and softly partitions its input space, dynamically learning how to do it and how to behave in each partition. In multiple layers of CMNs, the progressive-compositional development of such partitions can lead to instabilities or difficulties in quickly learning in an online manner. Preliminary investigations on fully connected networks, obtained by stacking multiple layers of CMNs, did not result in significant gain with respect to shallow architectures. Interestingly, when multiple layers are stacked, on average we observe a slight decrease in accuracy, both for our method and for the competitors. This finding confirms that larger models tend to perform better than deeper models for continual learning, due to exploding gradients leading to forgetting in the earliest layers, as discussed in Mirzadeh et al. (2022a;b).

## F    ADDITIONAL EXPERIMENTAL RESULTS

In Fig. 6 we report the upper-bound of the results presented in the main paper, obtained by selecting the best-performing model on the test data. Of course, these results are only intended to be a reference to understand the maximum performance each model could achieve, and not a way to make comparisons across different approaches. We remark that this is different from what we did in the main paper (Fig. 3), where we cross-validated the hyper-parameter values on the validation part of the stream and evaluated performance on the out-of-sample test sets.

Comparing Fig. 6 with Fig. 3, we notice that the CMN-based net is actually able to reach similar performance, thus being able to make the most out of the validation procedure. Since the validation set is limited to the first part of the streamed data, this results is very promising in terms of what can be achieved when working on longer streams with limited time-span dedicated to the validation of the parameters.

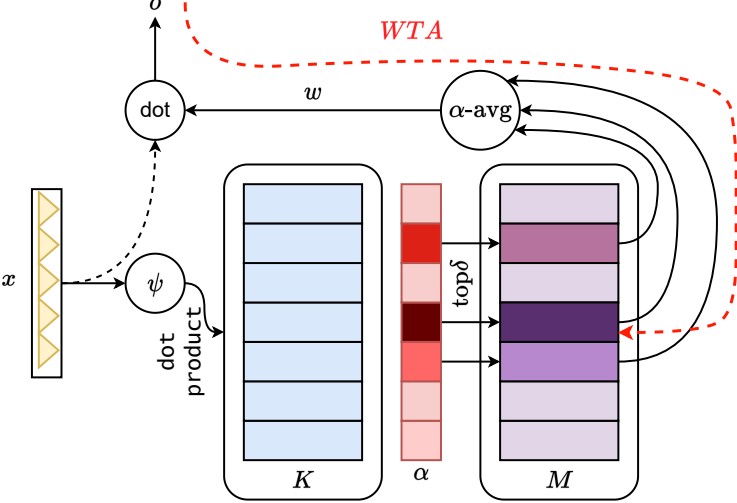

**Figure 7:** Larger instance of the contents of Fig. 1, for better readability.

