# OpenReview forum: "Continual Memory Neurons"
_ICLR.cc/2024/Conference — Submitted to ICLR 2024_

### Official Review · Reviewer_8Cfm · 2023-10-30

**Soundness:** 2 fair
**Presentation:** 1 poor
**Contribution:** 2 fair
**Rating:** 3
**Confidence:** 5

**Summary:**

The authors propose to develop a new type of neural-network neurons, termed Continual Memory Neurons (CMN), to address the catastrophic forgetting issue of lifelong learning. Specifically, learnable keys and memory units are introduced to parameterize the conventional weights of a neuron. Moreover, during lifelong learning, only top-δ activated keys and memory units are trained. Experiments on 2 simulated datasets, i.e., Modes and Moons, and 3 benchmark datasets, i.e., MNIST, CIFAR10, and NS-ImageNet, are conducted.

**Strengths:**

The presented techniques are likely novel, and the originality arises from the creative utilization of existing ideas for addressing new problems.

The research direction might be valuable for the lifelong learning community.

**Weaknesses:**

The writing should be significantly improved. The current manuscript is not easy to follow.

The presented techniques are not presented with convincing theoretical justifications. For example, what's the motivation for using Eq. (6) to update the keys K?

**Questions:**

Strong statements are made without convincing justification. For example, in Abstract, "...we are the first to follow a significantly novel route..."

In Eq. (4), why "set to 0 all the other excluded components?" What if all logits are less than 0?

How to select the hyperparameter $\delta$?

What are the insights in Section 2.1? How do the revealed connections with existing machine learning techniques contribute to the proposed CMN?

In the experiments, is a single data sample presented to the model at each time step? It seems in Figure 4 that each time step processes a new distribution (with many data samples)?

In Figure 5, it seems that CMN behaves differently on the 2-dimensional simulated datasets and the real-world imagenet dataset. Why?

---

> ### Author Response · Authors · 2023-11-19
> **Response to Reviewer 8Cfm**
>
> We thank the reviewer for carefully reviewing our paper. We also thank the reviewer for acknowledging the novelty and originality of the proposed ideas, as well as the importance of the field of research addressed in the paper.
>
> **Weakness1** We will proofread the paper and improve the writing, where necessary. If the reviewer would like to point out which sections of the paper were hard to follow we would appreciate the feedback so as to be able to better address the reviewer’s concern.
>
> **W2** Equation 6 is fully inspired by online K-Means. In fact, the WTA policy (as in K-Means) that we implement to update the memories in the neuron is what guarantees parameter isolation (thus avoiding forgetting), while also fostering forward transfer capabilities. The main idea is to let the network choose the $\delta$ most relevant memories using the similarity function defined in Equation 4, which will then be used in the update step. The remaining memories are going to be left untouched so as to prevent forgetting: if the memories are not relevant for the current sample, we do not want to update them to preserve the weights that were learnt for a different set of samples. At the same time updating $\delta$ memories rather than a single one can help to share information across similar tasks/distributions, which leads to forward transfer.
> Such considerations are confirmed by the experimental validation (e.g. Table 1), where forgetting is mitigated while forward transfer is present.
> In this update strategy, $\delta$ is a hyperparameter that can be adjusted depending on whether more isolation is preferable or not. We set $\delta=2$ for moons and modes and $\delta=5$ for ns-imagenet, as stated in the supplementary.
>
> **Q1** In our paper we propose a novel take on the continual learning problem by following a different route from prior work in the sense that we address the problem at the neuron level. The proposed Continual Memory Neuron is specifically designed to handle shifts in distributions and ingest ill-distributed data that can evolve continuously. Differently from prior works, we do not require any knowledge of task or distribution boundaries since weights are isolated in a data-driven fashion. In addition, we do not rely on rehearsal strategies that are commonly leveraged. Such strategies make use of non parametric memory buffers, require additional training steps to make a model recover forgotten capabilities and require to store samples based on some heuristics that again may be based on external knowledge about the current task or ground truth category. We believe that such research direction has not been investigated by other works.
> **We checked the paper for potential overstatements and removed them**.
>
> **Q2** We thank the reviewer for this comment and understand the confusion that arose from Equation 4. Actually, we perform the softmax activation only on the top-$\delta$ logits, setting to zero all the remaining elements in the softmax output. In this way we prevent unwanted behaviors when all logits are less than zero. **We clarified this in the updated manuscript**.
>
> **Q3**  As pointed out in the W2 answer above, $\delta$ is a hyperparameter that tunes the desired amount of parameter isolation. When the data exhibits lots of heterogeneous distribution shifts it is better to keep $\delta$ as low as possible, since this will harness the capability of isolating memories and specializing them for different distributions. At the same time, increasing $\delta$ can foster forward transfer. Note that when $\delta$ is set to be equal to the number of memories, all weights are updated for each training sample. This setting is likely the desired one in the presence of i.i.d. data. In our experiments, dealing with heavily biased data distributions, we set $\delta$ to 2 for moons and modes and 5 for ns-imagenet.
>
> **Q4** The insights in Section 2.1, as acknowledged by reviewer Lniw, should help to better frame the proposed approach in the state of the art, showing that CMNs lay their roots into well established approaches that involve memorization (Kernel Methods) and attention (Transformers). We believe that pointing out such a robust background will aid the understanding of the proposed approach as well as foster future research on the topic.
>
> **Q5** Indeed in the experiments a single sample is processed at a time. Figure 4 shows a snapshot of what the model is learning after having processed the last sample in each distribution. Actually, nor the training scheme nor the model are aware of task/distribution boundaries. Each subfigure in Figure 4 is thus showing the decision boundaries at different times, yet samples are processed one at a time. **We clarified this in the Figure caption**.
>
> **Q6**  In Fig. 5 (c,d) the main difference is due to the different dimensionality of the data. As expected, in fact, the cosine similarity/dot product is more appropriate in larger dimensional cases, such as the one of ImageNet.

---

### Official Review · Reviewer_9zgE · 2023-10-31

**Soundness:** 2 fair
**Presentation:** 3 good
**Contribution:** 2 fair
**Rating:** 5
**Confidence:** 3

**Summary:**

This paper proposes to change a regular weighted sum neuron to a more complex neuron where the weights are dynamically produced based on a set of key and memory vectors. The proposed neuron bears similarities to kernels and the attention mechanism. The authors then propose a learning mechanism based on winner-takes-all update as well as avoiding weak keys. Experiments on continual learning demonstrate the effectiveness of the proposed method.

**Strengths:**

(1) The proposed neuron is novel and insightful.

(2) The proposed learning algorithm is sensible.

(3) The experiment results demonstrate the effectiveness of the proposed method.

**Weaknesses:**

The main weakness is that the learning algorithm is ad hoc. It is not derived by minimizing a well-defined loss function.

The proposed neuron appears to be rather complex. Although the proposal is novel, the novelty is limited compared to the popular transformer model.

**Questions:**

Is the proposed method related to the mixture of experts?

Is it possible to introduce explicit latent variables to model the continual learning scenario?

---

> ### Author Response · Authors · 2023-11-19
> **Response to Reviewer 9zgE**
>
> We thank the reviewer for the insightful comments and for acknowledging the novelty and effectiveness of the proposed approach!
>
> **Weakness 1** Indeed, the update strategy for memory keys is ad hoc, but it is motivated by the typical requirements of continual learning settings. In fact, we made this choice being inspired by the online K-Means approach [Zhong, 2005], noticing that parameter isolation for task/distribution-wise learning shared some similarities with clustering approaches. In simple terms, we want similar neuron-inputs to be treated with the same weights. At the same time, it must be noted that weights are indeed learned via gradient-based optimization that minimizes the loss function. The combination of ad hoc updates for keys and gradient-based updates for memories gave us the ability to enforce the inductive bias of wanting to isolate parameters to deal with a time-shifting distribution in the data.
> However, we also compare our choice with a purely gradient-based approach (to update the keys), where both keys and values are updated with the task loss (Figure 5, $g_K$ variant of CMN). It can be seen how adopting this approach yields to far worse results than the proposed update strategy.
>
> **W. 2** Transformer models and CMNs share similarities only because they both use an attention mechanism. As discussed in Section 2.1, while transformers exploit attention to map the input into a new representation, establishing relative importance between parts of the inputs, CMNs instead use attention to determine which set of weights to use in a continual learning scenario. The effect of attention in Transformers is thus to obtain better representations to interpret the input, whereas the effect of attention in CMNs is to obtain parameter isolation and thus prevent forgetting. Furthermore CMNs are specifically tailored to deal with ill-distributed data. Interestingly, CMNs in principle could be used to improve transformers, replacing the learnable weight matrices that are present in Transformers.
> We thank the reviewer for sparking this interesting discussion, that we would like to deepen in future work.
>
>
> **Q. 1** As we discussed in Section 2.1 of the submitted paper (page 5, *“Ensemble Methods and ReLUs”* paragraph), Mixture of Experts (MoE) share similarities with the proposed Continual Memory Neurons.
> Indeed Sparse MoE can be described following the same notation as CMNs, where the $\alpha$ parameter weighing memories/experts is implemented by a learnable gating mechanism that selects only $K$ experts at once, propagating the gradients towards all of them. We believe that a fundamental component of the proposal is indeed the continual learning-oriented way in which $\alpha$ is defined, as well as the WTA approach that is not present in MoEs.
> Moreover, despite not being geared towards continual learning, Mixture of Experts shares with Continual Memory Neurons the goal of achieving input-based parameter isolation. However, two important differences have to be considered:
>
> - MoE is not thought for continual learning and especially not for incremental single-pass learning. This means that MoE, in its standard formulation, will easily be affected by catastrophic forgetting. Conversely, we proposed a novel training strategy, specifically tailored for handling continual streams of data, that makes CMN capable of overcoming forgetting. Such claims are supported by Table 1, where The MoE baseline is largely outperformed by CMN.
> - Architecture-wise, CMNs are still considerably different from MoE. In particular, MoE uses a gating strategy to select and weigh experts, thus enabling parameter isolation. At the same time, such gating mechanism (typically implemented as a multi layer perceptron or equipped with parameters learned by gradient descent), shares weights for all the inputs. In CMNs we use the input itself, followed by a mapping function $\psi$, to select the “experts” (memories), leveraging the key-value mechanism within the neuron itself. This fosters parameter isolation better than MoE, where the gating weights are shared and subjected to forgetting. Even when $K$ is shared across neurons in CMNs, isolation is better preserved, since keys and values are decoupled, differently than in MoE. Moreover, in principle we have the ability to adjust the granularity of the architecture's internal routes by not sharing the $K$ among neurons, discovering novel paths on demand.
>
> **Q. 2** CMNs are generic neuron models. The continual learning setting per-se can be defined in multiple custom ways.

---

### Official Review · Reviewer_Cr65 · 2023-10-31

**Soundness:** 1 poor
**Presentation:** 2 fair
**Contribution:** 4 excellent
**Rating:** 3
**Confidence:** 4

**Summary:**

The core contribution in this work is to introduce a mechanism for retrieval-augmented classification in a continual learning setting: It develops a key-value pair mechanism within NN layers so as to select the best memories per layer and utilize/update them.

I summarize the paper from a layerwise perspective as the mechanism until keys is shared across neurons.

The specific proposal is to replace every layer in the deep network from the following format:

A Linear Layer: $ f (x, W) = W^T.x$   where   $ x \in R^u , W \in R^{m \times u} $

to the proposed:

A Continual Linear Layer: $\forall_i (M_i^T$sparse-softmax$_{top-δ}{(γ · sim(ψ(x), K)})) x$

where $M_i$ is memory of every neuron in the layer, $M_i, K \in  R^{m \times u}$, $ψ$ is a function, while $γ$ is temperature and $δ$ is the sparsity value.

Notes:
- Every neuron has a distinct memory $M_i$
- Function $ψ$ chosen- identity
- Similarity metric ($sim$): cosine similarity for real experiments, RBF for 2D toy experiments
- $δ$ is a hyperparameter $\in {2, 5}$
- Extended to convolutional layers by considering each channel as a neuron

The design choice is in the selection/update mechanism for memories $M_i$:
- 1) Updating memories leads to forgetting: Ans) Select/update only the best memory $M_i[best]$ per layer to avoid interference (as keys are shared across neurons)
- 2) Avoid degenerate solutions (e.g. selecting the same key for all samples): Combination of the scramble and refresh key algorithm and online k-means like update to winning keys.

Can the authors correct my summary if there is something incorrect/missing? I found the paper quite hard to read, hence tried to give a slightly different perspective here to check whether I understood the mechanism correctly.
[Minor updates based on author feedback]

**Strengths:**

S1) **Tackles an important problem, quite convincing motivation [Critical]**

 The proposed continual memory neurons is a general and intuitively quite an effective mechanism to tackle catastrophic forgetting, and is quite distinct from current efforts in continual learning.

S2) **Well engineered [Critical]**

The approach was quite well engineered, the design-mechanisms both reduce computational overhead while trying to improve forgetting.

**Weaknesses:**

W1) **Missing References and Comparisons [Critical]**
- Nearly a page discusses connections to kernel methods, Transformers and RELUs-- however these seem quite non-central to the proposed mechanism.
- The proposed mechanism is closest to retrieval augmented continual learning works, which are surprisingly not discussed. Please see: https://github.com/hyintell/awesome-refreshing-llms for exhaustive references.

Why: Retrieval-augmented CL seems the closest to this work, as they introduce alternative mechanisms of continual memory neurons. It seems critical to understand how the proposed mechanism differs from existing proposals, as this papers claims (rightly) to be a general mechanism applicable across continual learning settings.
- I am specifically concerned about the mechanism of sparse selection + updates being better than alternative proposals.

W2) **Comparison between memory stored and images stored by MB [Critical]**
- The memory here is stored as weight matrices, which are inevitably smaller than storing images -- this in my opinion creates an unfair comparison.
- This is because CMNs can effectively store more samples than compared approaches, effectively achieving higher performance.

Furthermore: The memory constraint is by equalizing (1) the MB of storage, and (2) too low
- The proposed deep CMN networks (and comparisons) require higher GPU VRAM than the claimed available space on HDDs!

W3) **Too low absolute numbers on benchmarks to be meaningful, requires significantly better evaluation [Critical]**
I use MNIST-CI as an illustration to clearly show inadequacies in evaluation.
- Using a Nearest Class Mean [1] classifier on raw-pixels(!) achieves >85% accuracy on MNIST-CI, whereas reported SOTA here is 78%!
- For larger datasets, one needs some degree of features beyond raw pixels, but similarly, CIFAR10 performance of 27% is astonishingly low a bar to outperform.
- Benchmarks in the referenced continual-retrieval augmented transformers would be good candidates to compare performance.

Note that I think the proposed mechanism might be really useful, however the current evaluations seem too under-powered to verify benefits of CMNs.

[1] Mensink etal, Distance-Based Image Classification: Generalizing to new classes at near-zero cost, 2013

**Questions:**

Q1) **How do non-CMN methods perform with the RBF kernel in tthe moons/modes dataset? [Important]**
- Am I correct that the ψ(x) other than identity is only used in this scenario?
- What is the contribution of using the key-value pairing in CMN and RBF kernel for the task?
- Would equalizing the RBF kernel aspect significantly affect the performance gain of CMN method?

Q2) **Why are the results varying so much in MNIST-CI? [Important]**
- It is very strange to see 21.0 for ER-Random, 23.2 for MoE and 14.3 for GDumb but 70.3 for ER-Reservoir-Imbalanced.
- Reservoir-Imbalanced sampling (Chrysakis & Moens, 2020) approximates random sampling as there is no imbalance to correct so they should have identical performance.

Overall, 14.3/21.0/23.2 on MNIST-CI seems suspiciously low performance despite the memory size being 100!

---

> ### Author Response · Authors · 2023-11-19
> **Response to Reviewer Cr65 1/2**
>
> We thank the Reviewer for the work put in reviewing our paper. We do believe there are several misunderstandings in the review that we try to solve in what follows.
> The summary provided by the Reviewer catches many of the key aspects of the proposed method, but it includes several misunderstandings that likely hindered the Reviewer's comprehension of the paper. We hope that the following clarification could help the Reviewer in better grasping the model details.
>
>  - From the paper:
>    >  “ The function $\psi$ is a fixed transformation that maps $x$ toward a customizable space in which it might be easier to compute similarities… but nothing prevents $\psi$ to be the identity function \footnote{That is what we did in our experiments. There is room for investigating the way $\psi$ could be defined---beyond the scope of this paper.}.
>
> - We remark that $\psi$ is not a kernel, as the Reviewer misunderstood, and it was set to the identity function, for simplicity. It can be further customized, of course. Please, notice that the RBF function mentioned in the paper is an example of implementation of the similarity measure  $\mathrm{sim}$ that compares keys with neuron input. Other possible custom choices can be considered (in the paper we also mention cosine similarity and dot product -- see the following).
>
> - In the Reviewer’s summary, the Review reported $\mathrm{sim}$ to be always the cosine similarity, but that is not correct. The similarity function $\mathrm{sim}$ is not always the cosine similarity. Please refer to the last paragraph of Section 2 of the submitted paper, where we remarked that:
> > “There exist several different ways of implementing $\mathrm{sim}$, such as the dot product (scaled by the square root of  the key size), the cosine similarity, RBF kernels $\left[ \exp\left(-\frac{1}{2\tilde{\sigma}}\| \psi(x) - K_i \|^2 \right)\right]_{i=1}^{m}$, being $\| \cdot \|$ the $L_2$ norm, and others. In our experience, we used the RBF implementation in $2$-dim cases, while the the cosine similarity in all the other experiments of this paper.”
>
> -  In fact, please also notice that in Figure 5 we tested  $\mathrm{sim}$  to be either cosine similarity, scaled dot product or RBF.
>
> - We do not set $\delta$ to one, but instead we use $\delta=2$ for moons and modes and $\delta=5$ for **ns-imagenet** (in which there is more variability), as stated in the *“Hyper-parameters”* section of the supplementary material (appendices). However, $\delta$ is a hyperparameter that can be adjusted depending on whether more isolation is preferable or not. Concerning design choice 2 in reviewer’s summary, the update does not concern the best memory but the best $\delta$ memories.
>
> **Weakness 1**  Since we are proposing a novel neuron model, one might wonder how it is computationally different from what happens in popular neural architectures that share some of the intuitions that are also present in CMNs, that is why we investigated the connection with them. For example, Kernel Methods, in their dual form, learn function that depend on all the seen training examples, thus they do memorize all the data, and we found it important to see how CMNs, being them memory-oriented neurons, are related to them (of course, CMNs do not store all the data). Moreover, since we are building on key-value functions, the connection with Transformers seems appropriate. We remark that other Reviewers (Lniw) appreciated this discussion.
> The reviewer emphasizes the similarity between our work and retrieval-augmented continual learning, referring to a list of recent works on LLMs where retrieval is interpreted as access to external sources of information (e.g., search engines). This is entirely different from what we do. We propose an enhanced version of a neuron that learns different set of weights to handle non-stationary data distributions during training. In any case, our neural architectures are entirely self-contained without any proper ‘retrieve’ operation. The only thing in our model which is vaguely connected with retrieval is the attention strategy to access the memories.
> In a sense, every attention-oriented model can be intended as retrieving keys from a storage, thus we do see  the connection that the Reviewer is pushing, but we find it to be way too weak/generic. CMNs work at the lowest possible abstraction level, and they could be paired with any other existing CL approaches, including Retrieval-augmented CL, experience replay, … We ask the Reviewer to better point out which methods we should discuss in the ‘related works’ section, in order to provide an appropriate answer.

---

> ### Author Response · Authors · 2023-11-19
> **Response to Reviewer Cr65 2/2**
>
> **W. 2** The Reviewer is remarking that CMNs store *“more examples”* than other methods in the *“weight matrices”* that are smaller than storing images. We compared our approach and exemplar based competitors from the computational and memory point of view, and we proposed an in depth analysis in  Appendix C of the submitted paper geared toward a fair comparison. Storing data in a more efficient way is indeed an important feature of CMNs. We believe that having the ability to store more information with the same amount of memory is a crucial characteristic in the online continual learning setting. We do not understand the short comment on the GPU memory. Also mini-batches of rehearsal buffer are loaded into GPU VRAM when processing mini-batches, and our comparisons were built in a way in which such memory consumption when comparing CMNs and standard neurons  is the same.  The Reviewer refers to *“too low”* memory constraints, but we remark that we deal with an online learning scenario where frugal data storage is an extremely important requirement.
>
> **W. 3**  As usual, numbers must be evaluated in the considered experimental setup, that is the one of a continual online learning scenario. We really do not understand the comments about the comparison with [1], which is about a metric-learning approach, assessed on a completely different setting and dataset. The results we provided include a large number of comparisons with SOTA approaches. We believe that experimental results should not be superficially dismissed just because ‘accuracy looks too low’ and we stress that reproducibility of experiments is possible since we shared the code. We recall we are in a continual online setting (1 example at a time, no epochs, no batches,…), that is way more extreme that what is explored in several other papers! The choice on the simple neural architecture is what yields the performance we reported in CIFAR10 - again, this is continual online learning! The learning setting is indeed challenging and it is a precious testbed for CMNs and for comparisons. Recall (see the paper) that we validated the model parameters in a realistic setting in which a small proportion of the stream is exploited, that makes the setting more appropriate for comparisons and also more tough.
>
> *[1] Mensink etal, Distance-Based Image Classification: Generalizing to new classes at near-zero cost, 2013*
>
> **Q. 1** We remark that $\psi$ is not a kernel, but a fixed transformation that maps $x$ toward a customizable space in which it might be easier to compute similarities. We  set $\psi$ to the identity function in all the experiments, for simplicity. Indeed, both CMNs and non-CMN methods process the same data $\psi(x)=x$.   Conversely, the RBF kernel is one of the possible implementations for the similarity function  $\mathrm{sim}$ that drives comparison with keys (see the first part of our response ).
> We do not understand the comparisons that the Reviewer is indicating in the “questions” box. Changing the variance of the kernel will not affect the WTA procedures, but it would change the way the winning key is updated.
>
> **Q. 2**  Reservoir-based management of the buffer reduces the probability of adding a sample to it as long as time passes, and has been proven to be very effective in the Continual Online Learning scenario [2]. In bare random-management of the buffer, it is more likely to have replacements (that leads to a buffer containing a lot of recent samples), that is why performances are different. See [2] for further details on the difference between the two algorithms.
>
> *[2] Mai, Zheda, et al. "Online continual learning in image classification: An empirical survey." Neurocomputing 469 (2022): 28-51.*

---

### Official Review · Reviewer_Lniw · 2023-11-04

**Soundness:** 3 good
**Presentation:** 3 good
**Contribution:** 3 good
**Rating:** 6
**Confidence:** 4

**Summary:**

The paper proposes continual memory neurons (CMNs) which is a key-value attention-based memory module that distributes the past memory into slots. A winner-take-all retrieval and forgetting mechanism is also proposed. When a single layer of CMN can achieve superior performance on online continual learning compared to baselines that use example buffers and replay.

**Strengths:**

- It is good that the authors discuss the relation of CMN to classic neurons, transformer networks and kernel machines.
- Experiments show that the network is significantly better at online continual learning compared to baselines that use example buffers and replay.

**Weaknesses:**

- It seems that in the experiments the authors have only tried with a single layer of CMN, but the naming of “neuron” and the methodology suggests that it can be applied more widely across layers. It would be better to showcase the general applicability into multi layer CMNs. A single CMN can be achieved with a more standard online clustering algorithm.
- To continue with the previous point, a concern with applying CMN to multiple layers is that when the representations are not fully trained or in early layers, committing to the winner memory slot may hinder the learning progress, but more studies would be needed.
- The memory unit has resemblance to Ren et al. (2021) so it would be great to discuss its relations in the paper. Both works use a slot-based memory module based on input similarity. Both recycle the least used entry and create a new entry if the match strength is below a threshold.
- Hyperparameters selection of CMN seems like a burden. It would be great to show a list of hyperparameters and their optimal range for each task, and discuss how sensitive these values are.

References:
Ren, M., Scott, T. R., Iuzzolino, M. L., Mozer, M. C., & Zemel, R. (2021). Online unsupervised learning of visual representations and categories. arXiv preprint arXiv:2109.05675.

**Questions:**

N/A

---

> ### Author Response · Authors · 2023-11-19
> **Response to Reviewer Lniw**
>
> We thank the Reviewer for the attention reserved to our work and for having appreciated our discussion on the connections of our computational scheme with classic neurons, kernel machines, etc.
>
> **Weakness 1**   As the Reviewer correctly suggests, we devised CMNs to be general neural components that could be plugged into any arbitrary architecture. We achieved impressive results exploiting single layered CMNs architecture, even surpassing the performance of models equipped with several layers of standard neurons, which require, in addition, to leverage rehearsal buffers in order to be competitive in the continual learning scenario – that is not our case. Thus, in principle, the proposed CMNs are general enough to be plugged into any neural architecture, and they can be paired with existing approaches for continual learning. However, their interactions in deep networks with several layers, or the way they behave with other learning approaches still require specific and detailed investigations. For example, we focused on networks in which the input of the CMNs was not affected by the learning dynamics, both when using backbones with several layers or when directly classifying data in their original representations. Each CMN virtually and softly partitions its input space, dynamically learning how to do it and how to behave in each partition. In multiple layers of CMNs, the progressive-compositional development of such partitions can lead to instabilities or difficulties in quickly learning in an online manner.
> This will be the subject of our future work. We remark that many other works in the CL scenario propose to leverage pretrained/freezed backbones and to learn on top succinct memory systems that exploit the underlying knowledge by selecting the most appropriate one depending on the current sample [a]. We believe that our proposal is per-se an already relevant contribution in this direction, as evidenced by the Imagenet-NS experiment.
> Moreover, by stacking multiple layers of CMNs we did not observe any significant gain. Interestingly, when multiple layers are stacked, on average we observe a slight decrease in accuracy, both for our method and for the competitors. This finding confirms that larger models tend to perform better than deeper models for continual learning, due to exploding gradients leading to forgetting in the earliest layers, as discussed in [b, c]. We added these considerations in the revised paper (please see Appendix E).
>
> *[a] Wang, Zifeng, et al. "Learning to prompt for continual learning."  IEEE/CVF CVPR 2022.*
>
> *[b] Mirzadeh, S. I. et al. 2022. Wide neural networks forget less catastrophically. In ICML  2022*
>
> *[b] Mirzadeh, S. I. et al. 2022. Architecture matters in continual learning.  arXiv:2202.00275.*
>
>
> **W.  2**   We indeed agree with the Reviewer that more studies are needed in the case of multiple layers.  Please also refer to our response to W1. We do believe the novelty and the experimental quality of this proposal are enough to yield a significant impact in the context of a conference.
>
> **W.  3**  Thanks for this interesting suggestion! We referenced the paper in the updated submission, due to some analogies with a few components that are also present in what we propose, even if the two proposals are indeed overall different.
> Indeed, Ren et al. propose a prototype-based memory network with a control component that determines when to form a new class prototype. This is done via a probabilistic clustering module that is jointly learned with the representation encoder. Cluster assignment updates and parameters estimates are achieved via the EM-algorithm based on the input-prototypes similarity, in a streaming setting. The cluster formation is guided by a self-supervised contrastive loss that encourages different views of the same image to be assigned to the same prototype, in addition to other loss terms that are needed to guarantee the formation of new clusters. Least used clusters are removed based on thresholds.
> Conversely, we propose a novel neuron model leveraging a decoupled key-value mechanism within the neuron itself. While keys are updated via an online clustering method on the input space, top-$\delta$ memories are learned via the backpropagated gradients computed on the current loss function without the need of ad-hoc loss terms. Every CMN has its own independent memory bank, and CMNs can be combined into layers.
>
> **W.  4**  We thank the Reviewer for this feedback. In the supplementary materials (appendices) we reported the hyperparameter grids and the best selected parameter for the main results. We also investigated the role of key design choices in the ablation study (Fig. 5). In our experience, their impact was prevalent, so we found it more meaningful to provide this kind of information rather than a fine-grained exploration of the hyperparameter values. Of course, there is always room to investigate deeply the sensitivity to each hyperparameter.

---

### Meta-Review · Area_Chair_XQdA · 2023-12-04

**Metareview:**

The manuscript is reviewed by four expert reviews. Three of the argued for rejection and one of them weakly argued for acceptance. Authors responded to the reviews. I carefully read the paper, reviews and the authors' response. I believe the paper should be rejected in its current from since:

- It is lacking clarity: The paper is very hard to read. Worryingly multiple reviewers misunderstood the significant details about the paper. Considering the paper is not even accessible to experts in online continual learning, its accessibility to the general ICLR audience is very unlikely.

- Experimental setup: Authors do not follow the standard settings. Hence, the reported baseline numbers are all self-reported instead of using already published numbers. From the current experimental section, I do not have a good sense of performance of the proposed method with respect to other OCL methods.

In summary, the proposed method is not clear and is not well evaluated. I strongly recommend authors to fix these issues and re-submit.

**Justification For Why Not Higher Score:**

The paper is not ready for publication. It is not clear and lacks a rigourous empirical study.

**Justification For Why Not Lower Score:**

N/A

---

### Decision · Program_Chairs · 2024-01-16

Reject